# The Multifaceted Role of Confocal Laser Endomicroscopy in Head and Neck Surgery: Oncologic and Functional Insights

**DOI:** 10.3390/diagnostics13193081

**Published:** 2023-09-28

**Authors:** Nina Wenda, Kai Fruth, Annette Fisseler-Eckhoff, Jan Gosepath

**Affiliations:** 1Department of Otolaryngology, Head and Neck Surgery, Helios HSK Wiesbaden, 65199 Wiesbaden, Germany; kai.fruth@web.de (K.F.); jan.gosepath@helios-gesundheit.de (J.G.); 2Department of Pathology, Helios HSK Wiesbaden, 65199 Wiesbaden, Germany; annette.fisseler-eckhoff@helios-gesundheit.de

**Keywords:** real time imaging in head and neck surgery, CLE, allergic rhinitis, nasal provocation testing

## Abstract

(1) Background: Confocal laser endomicroscopy (CLE) has emerged as a transformative tool in head and neck surgery, with applications spanning oncologic insights and functional evaluations. This study delves into CLE’s potential in these domains. (2) Methods: We performed CLE in head and neck oncologic surgery, focusing on tumor margin identification and precise resection. We also employed CLE for functional assessment in allergic rhinitis, observing real-time mucosal changes during nasal provocation testing. (3) Results: In oncologic surgery, CLE enabled real-time visualization of tumor margins and cellular patterns, aiding resection decisions. In allergic rhinitis assessment, CLE captured dynamic morphological alterations upon allergen exposure, enhancing understanding of mucosal reactions. (4) Conclusions: The integration of CLE with evolving technologies such as deep learning and AI holds promise for enhanced diagnostic accuracy. This study underscores CLE’s expansive potential, highlighting its role in guiding surgical choices and illuminating inflammatory processes in the head and neck.

## 1. Introduction

Advancements in real-time imaging continue to revolutionize various fields of surgery, and one such innovation is confocal laser endomicroscopy (CLE). Originally developed for the field of gastroenterology, CLE has found valuable applications in head and neck surgery [1,2,3].

Confocal laser endomicroscopy is a high-resolution imaging technique that provides real-time microscopic images of mucosal tissue. The key feature of CLE is its ability to capture cellular and subcellular details, enabling surgeons to visualize tissue structures at a microscopic level during ongoing endoscopy without the need for traditional histopathological processing [4].

In head and neck oncologic surgery, precise identification of tumor margins is crucial for achieving optimal outcomes in terms of complete tumor resection and maximum preservation of healthy tissue at the same time.

The ability to assess tumor margins dynamically using CLE curtails the risk of leaving behind residual cancerous tissue [5].

Beyond oncologic surgery, CLE embarks on an experimental journey toward functionally assessing endonasal mucosa in allergic rhinitis.

As a real-time gaze into inflammation, CLE enables observing mucosal changes characteristic of allergic rhinitis, facilitating accurate evaluation of the inflammatory response [6].

This article delves into the dual role of CLE in the head and neck, highlighting its significance in oncologic surgery and its potential in functionally assessing endonasal inflammatory processes.

## 2. Materials and Methods

This study aimed to investigate the application of CLE in both oncologic surgery involving various locales of head and neck tumors and during nasal provocation testing for patients with known allergic rhinitis. Patients were enrolled from a tertiary medical center in Germany. The study was conducted following the principles outlined in the Declaration of Helsinki. Approval was obtained from the institutional review board (IRB) and the ethics committee of Hesse, Germany (FF 146/2017). Written informed consent was obtained from all participants before their inclusion in the study.

In CLE, a low-power laser is focused onto a single point within a microscopic field of view. Utilizing the same optical lens for both condensation and objective functions, the optical path is folded, aligning the point of laser illumination precisely with the point of detection embedded within the specimen. This strategic alignment between illumination and detection, existing on the same focal plane, substantiates the term “confocal”.

The acquisition and analysis of signals emitted from the illuminated point constitutes the process in this technological approach. These signals are systematically quantified, and their intensity is subsequently translated into grayscale representations, visually rendering the intricate structure of microscopic tissue.

Over recent years, the progressive advancement of this technology has led to the integration of a miniaturized laser scanner within the distal portion of a conventional flexible video endoscope. This miniaturization facilitates the visualization of mucosal intricacies at subcellular resolutions during the course of endoscopy.

The probe-based system (pCLE, Cellvizio^®^ Endomicroscopy System, Mauna Kea Technologies, Paris, France) is comprised of a 1.6 mm flexible miniprobe. This miniprobe offers a field of view encompassing a diameter of 240 µm, enabling observations within a confocal depth range of 55–65 µm with a resolution of 1 µm.

The miniprobe units are adaptable, serving either as independent tools or adeptly inserted through the working channel of a diverse array of endoscopes for clinical applications. Following the intravenous administration of 2.5 mL of fluorescein (100 mg/mL) as a contrast agent, confocal assessment is possible within approximately 30 s.

### 2.1. Oncologic Surgery

The study cohort comprised patients diagnosed with various head and neck malignancies, including squamous cell carcinoma of distinct locales such as the nasal vestibule, soft palate and lateral tongue. These patients were chosen based on their confirmed malignancy diagnosis and scheduled for surgical resection. Figure 1 exemplarily shows the clinical setting of the use of CLE in transoral resection of an oropharyngeal carcinoma. Prior to resection, CLE was employed to scrutinize the margins of the lesions, aiming to visualize the extent of the tumor. Following resection, CLE was once again utilized to assess the surgical site, specifically targeting the presence of any residual tumorous cells in the area of the resection margin.

To further validate the CLE findings, biopsies were obtained from distinct locations for histopathological evaluation. These locations included the center of the tumor, the tumor margin, and the surrounding healthy mucosa. The biopsy sites were determined in line with the insights gleaned from CLE imaging and the pre-determined resection approach by experienced head and neck surgeons.

Images generated through CLE were then presented to two independent pathologists for comparative analysis. This analysis encompassed the identification of cellular characteristics, quality evaluation of CLE images, and the degree of congruence with traditional histopathological hematoxylin and eosin (H&E) staining. Comparative analyses between CLE and histopathological findings were conducted qualitatively, with a focus on concordance between the two modalities.

### 2.2. Nasal Provocation Testing in Allergic Rhinitis

We examined four individuals between the ages of 21 and 31. Among these participants, two displayed sensitization to birch pollen (IgE class III and V), while one individual showcased sensitization to house dust mites (IgE class III). Another participant exhibited sensitization to timothy grass (IgE class IV).

Subsequent to the intravenous administration of fluorescein, the mucosa of the inferior turbinate was examined with the confocal laser probe to assess the baseline cellular configuration.

Following this initial examination, nasal provocation testing was executed. Various allergens were employed for this purpose: birch pollen (0.04 mL, equivalent to 0.015 HEP [histamine equivalent prick]), dermatophagoides pteronyssinus (0.04 mL, equivalent to 0.05 HEP), and timothy grass (0.04 mL, equivalent to 0.015 HEP; LETI Pharma, Ismaning, Germany). Through this testing, we scrutinized the ensuing mucosal reactions, considering concurrent symptoms and the temporal progression. This observation was accomplished over a span of 5 min for each participant. Figure 2 displays the performance of CLE at the beginning of the procedure.

## 3. Results

### 3.1. Oncologic Surgery

The implementation of CLE proved to be consistently feasible across the examined spectrum of head and neck tumors, with successful initiation approximately 30 s post-intravenous administration of fluorescein.

We did not observe any unexpected side effects due to the administration of the contrast medium in any of the patients examined. The examination with CLE prolonged the respective procedure by approximately 10 min.

While the quality of images varied contingent upon factors such as tumor site, tissue vascularization and tissue vulnerability, we were able to demonstrate the capacity to capture images of commendable quality across all regions under scrutiny.

Delving into the examination of healthy mucosal zones surrounding the tumors, the results unveiled consistent patterns. The observed configuration remained uniform, accompanied by well-defined cell margins and homogenous uptake of fluorescein.

Within tumor regions, the landscape underwent transformation. The cellular architecture exhibited disarray, diverging from the ordered structure noted in healthy tissue. Concomitantly, capillaries within tumor sites exhibited enlargement, contributing to the observed irregular cellular patterns. The uptake of fluorescein within tumor areas appeared uneven, generating an intricate mosaic of cellular imagery. Table 1 summarizes the morphological characteristics of healthy and neoplastic tissue in CLE imaging.

The findings from CLE were corroborated through histopathological assessments. Expert pathologists successfully recognized the cellular features described through CLE in corresponding histopathological sections. Furthermore, a high congruence emerged when comparing CLE and histopathological images. The juxtaposition of CLE images and the corresponding histopathological section is presented in Figure 3.

#### 3.1.1. Endonasal Squamous Cell Carcinoma (ESCC)

The use of CLE in endonasal malignancies, particularly those involving the nasal vestibule and the nasal cavity, was highly implementable. Here, the positioning of the laser probe was easily feasible, allowing the acquisition of images of satisfactory quality. Moreover, malignancies of the paranasal sinuses and skull base similarly lent themselves to CLE examination; thus, the positioning of the probe was more demanding in this area. The full range of tumor margins could be meticulously examined both pre- and post-resection, furnishing comprehensive insights into the extent of malignancies.

The maneuvering of the CLE probe through the intricate anatomical pathways of the nose and paranasal sinuses was well executable. Notably, with the assistance of rigid instruments, successful probe insertion within constrained regions such as the maxillary sinus or frontal recess was achievable with generation of satisfying image quality. Figure 4 portrays the examination of an ESCC of the right nasal vestibule.

#### 3.1.2. Oropharyngeal Squamous Cell Carcinoma (OPSCC)

Within the oropharynx, certain locales exhibited heightened suitability for CLE examination. Regions such as the tonsils and the hard and soft palate were notably accessible, fostering facile imaging. Conversely, challenges arose when examining the base of the tongue due to its intricate anatomy characterized by folds and furrows. Here, precise probe positioning demanded additional time and effort to achieve satisfactory image quality. Figure 5 shows the examination of an OPSCC of the soft palate.

#### 3.1.3. Oral Cavity Squamous Cell Carcinoma (OCSCC)

In the oral cavity, specific sites emerged as apt locales for CLE examination. Notably, the floor of the mouth and the tongue demonstrated a high degree of suitability. Examination of the lateral aspect of the tongue was comparatively straightforward, whereas the back of the tongue posed greater complexity due to extended keratinization. Figure 6 displays the examination of an OCSCC of the lateral tongue.

### 3.2. Functional Assessment of Allergic Rhinitis

As in the investigation of endonasal carcinomas, the application of CLE in the nasal cavity in the context of nasal provocation testing was shown to be feasible and images of high-quality could be achieved.

The occurrence of artifacts primarily stemmed from two sources: unstable positioning of the laser probe and mucus adhesion. Notably, the comprehensive examination process was efficiently accomplished within a time frame that did not exceed 10 min.

Before the initiation of allergen exposure, the nasal mucosa exhibited regular patterns with narrow intercellular cleavages, reflecting a baseline state of cellular architecture.

Subsequently, upon application of the respective antigen approximately 30 s to 1 min thereafter, a progression of dynamic alterations manifested. These included a gradual widening of intercellular gaps, the emergence of perivascular edema and the conspicuous dilation of capillaries. Over the subsequent time interval, a consistent augmentation of intracellular volume was observed within the superficial epithelial layer.

At the acme of the allergic response, typically manifesting within 3 to 5 min post-antigen application, a remarkable cascade transpired. A massive disintegration of numerous cells materialized, contributing to an augmented endonasal secretion—a manifestation that symbolized the culmination of the allergic reaction’s intensity.

Figure 7 displays representative endomicroscopic images of these morphological changes before and after allergen challenge.

## 4. Discussion

### 4.1. CLE in Functional Assessment of Allergic Rhinitis

In the context of contemporary medical landscapes, allergic rhinitis prevails as one of the most pervasive allergic conditions, exhibiting a current lifetime prevalence exceeding 20% [7]. While substantial strides have been made in unraveling the pathomechanisms of allergic rhinitis, the realm of comprehending cellular, vascular and immunological shifts, alongside their intricate interplay, remains an ongoing challenge for researchers vested in this field [8].

At the heart of mucosal pathophysiology lie IgE-mediated reactions that set in motion cascades of vascular and glandular responses. The cornerstone of these responses lies in the release of inflammatory mediators, including histamine, bradykinin and leukotrienes. This release predominantly emanates from mast cells and basophil granulocytes [9]. As a consequence, a constellation of vascular effects takes shape, characterized by vasodilation, transudation and perivascular edema. Simultaneously, submucosal secretory glands undergo heightened secretion [10]. Herein, the excitation of sensory nerves plays a pivotal role, ushering in the release of neuropeptides that, in turn, potentiate these multifaceted mechanisms [11]. The culmination of these intricate vascular, glandular and neural reactions collectively manifests in the array of clinical presentations such as nasal obstruction, rhinorrhea and sneezing [12]. To our knowledge, our group was the first to successfully perform a real time visualization of these complex processes using CLE [6].

Upon initiation of nasal provocation, our observations unveiled the initial emergence of vasodilation, coupled with the concurrent development of perivascular edema. These early events stem from the loss of intercellular junctions, signifying the perturbation of local barriers within the nasal mucosa. The phenomenon of gap formation and local barrier dysfunction bears significance beyond the realm of allergic rhinitis, echoing throughout acute and chronic inflammatory diseases. The utilization of CLE has enabled comparable insights in divergent clinical domains. Notably, Lim et al. harnessed CLE to detect epithelial damage and the loss of barriers within the terminal ileum of individuals afflicted with inflammatory bowel disease [13]. The use of CLE could likewise reveal an increased duodenal permeability in patients with acute pancreatitis [14]. Moreover, our investigations unveiled a notable augmentation in intracellular volume, culminating in eruptions akin to volcanic phenomena from individual superficial cells. Using CLE, analogous effects can be found within the realm of food allergies. Following exposure to food antigens, the apical epithelium of intestinal villi has been documented to undergo disruption, leading to the shedding of cells. This process engenders breaches within the mucosal barrier, thereby facilitating the influx of fluorescein into the luminal space [15].

The insights gleaned from the utilization of CLE in allergic rhinitis have the potential to enrich the diagnostic toolkit for forthcoming investigations of inflammatory processes. This augmentation is poised to illuminate various fronts, including the exploration of prevalent disparities often encountered between clinical manifestations and the outcomes of both skin and in vitro testing. Equally imperative will be the exploration of its diagnostic efficacy in chronic inflammatory conditions like chronic rhinosinusitis.

### 4.2. CLE in Head and Neck Surgery

CLE emerges as a promising tool in head and neck surgery, bestowing the ability to furnish real-time insights into tumor dynamics and offering intricate evaluations of tumor margins. This capability carries positive implications for patient safety by enabling precise delineation of tumor boundaries, thereby potentially reducing the need for unnecessary resections. In the context of the important anatomical structures in the immediate vicinity within the head and neck region, accurate margin assessment assumes paramount significance, safeguarding functional outcomes. The clinical implications of judicious resection extend beyond classical factors such as progression-free and overall survival. The preservation of functional abilities, such as swallowing, vocalization and mitigating cosmetic repercussions can profoundly impact a patient’s quality of life [16]. A precise assessment facilitated by CLE holds promise in curbing the extent of resections while optimizing both oncologic and functional outcomes.

Numerous optical imaging techniques, including narrow-band imaging, fluorescence endoscopy and optical coherence tomography have been proposed for their potential to enhance the assessment of mucosal lesions in the head and neck using white light [17,18,19,20].

In 2012, our group was one of the first to develop a systematic intraoperative evaluation of CLE in the oropharynx [21]. After further development and miniaturization of the laser probe, we were able to expand the scope of application and apply the knowledge gained in the process to evaluate neoplasms of the nose and the paranasal sinuses [22,23].

CLE is only possible using fluorescence contrast agents. Intravenously applied fluorescein sodium (2.5 mL of 100 mg/mL fluorescein 10%) distributes throughout the entire mucosa with a strong contrast within the connective tissue and the capillary network. Fluorescein’s mechanism of action is tied to its binding with serum albumin, facilitating its robust distribution. Unbound fluorescein molecules traverse capillaries to infiltrate the tissue, conspicuously illuminating the extracellular matrix. This dynamic interplay, occurring within seconds post-injection, empowers the rapid initiation of CLE examinations [24].

Intravenous fluorescein stands as a well-established and clinically embraced agent, widely adopted in ophthalmology. Its extensive history of safe clinical usage has cemented its reputation as a nontoxic option. Most common side effects related to fluorescein administration, such as vomiting and nausea, are transient and minor. More severe side effects, like vasovagal response, cardiac or respiratory response are extremely rare [25].

Of course, CLE has technical as well as user-related limitations and challenges. Effective manipulation of the laser probe necessitates a learning curve, underlining the significance of surgeon familiarity. Notably, achieving the correct contact pressure and angle for optimal image quality can be challenging, varying based on anatomical regions. As CLE technology advances, innovations should tread the fine line between stabilizing the probe and maintaining its inherent flexibility. This balance is particularly crucial in regions like the paranasal sinuses, where maneuverability within the narrow anatomy remains indispensable. Supporting tools should be designed to enhance probe stability at the tip of the probe without compromising overall flexibility.

Likewise, the interpretation of CLE images is subject to a learning curve. In the beginning, it is crucial to note that the generated grayscale images depict a horizontal plane of the examined mucosa. Accordingly, images are perpendicularly displayed compared to the classical histopathological sections. This distinctive perspective requires mindfulness, particularly for histopathologists seeking concordance across modalities.

The assessment of sensitivity and specificity of CLE differs remarkably across investigative groups. Abacci et al. achieved a sensitivity and specificity of CLE imaging of 73.2–75% and 30–57.4%, respectively [26]. However, one omission in their study lies in the absence of information regarding the level of experience possessed by their surgeons and pathologists in utilizing CLE. Additionally, their choice of a topically applied contrast agent, as opposed to the intravenously applied fluorescein, could have potentially influenced image quality. In order to improve the evaluation of CLE images, the group of Sievert et al. devised a scoring system encompassing key factors like tissue homogeneity, cell size, borders and clusters, capillary loops and the nucleus/cytoplasm ratio in neoplasms of the pharynx and larynx [27]. Presenting these images to both CLE experts and non-experts, they revealed distinct outcomes. CLE experts achieved striking accuracy, sensitivity and specificity of 90.8%, 95.1% and 86.4%. Similarly, CLE non-experts demonstrated respectable scores of 86.2%, 86.4% and 86.1%. The marked improvement in both accuracy and sensitivity underscores the pivotal role of expertise in this technology and its resultant images. The group of Sievert et al. also succeeded in transferring the data obtained in the pharynx and larynx to tumors of the oral cavity and therefore validate the scoring system [28]. Accordingly, a transfer to other regions of the head and neck such as the nose and paranasal sinuses also seems plausible.

A meta-analysis undertaken by Sethi et al. included six studies, encompassing a total of 361 lesions across 213 patients with oral squamous cell carcinoma (OSCC) examined with CLE. The pooled outcomes for sensitivity and specificity were notably robust, standing at 95% (95% CI, 92–97%, I^2^ = 77.5%) and 93% (95% CI, 90–95%; I^2^ = 68.6%) [29]. This consolidated analysis reaffirms the capacity of CLE to deliver high-fidelity diagnostic outcomes in the context of OSCC.

In the era of burgeoning deep learning systems and artificial intelligence (AI), the potential for augmenting CLE-generated data evaluation looms large. Machine learning approaches have demonstrated remarkable efficacy in diverse medical domains. Guleria et al. installed a deep learning model to detect dysplasia in patients with Barret’s esophagus (BE) with CLE and histopathological sections. They achieved high diagnostic accuracy for both CLE-based and histopathologic diagnosis of esophageal dysplasia and its precursors, similar to human accuracy [30]. Lee et al. also advocate the use of deep learning models in evaluation of pancreatic cystic lesions (PCL) by CLE [31]. Aubreville et al. presented a significantly improved overall performance of 94,8% using a model for deep learning-based detection of motion artifacts in CLE images [32].

However, the profound impact of AI must be underpinned by seasoned clinical experience. Expertise in probe manipulation and adeptness in handling challenging anatomical sites remain the bedrock. As deep learning and AI ascend, their synergy with experienced clinicians and surgeons can become the fulcrum for optimizing CLE image evaluation and decision-making.

## 5. Conclusions

The potential of CLE in the head and neck is extensive. The convergence of CLE with emerging technologies like deep learning and AI holds great promise for enhancing its accuracy. Larger studies should solidify CLE’s value in guiding surgical decisions and aiding functional evaluations in inflammatory conditions.

## Figures and Tables

**Figure 1 diagnostics-13-03081-f001:**
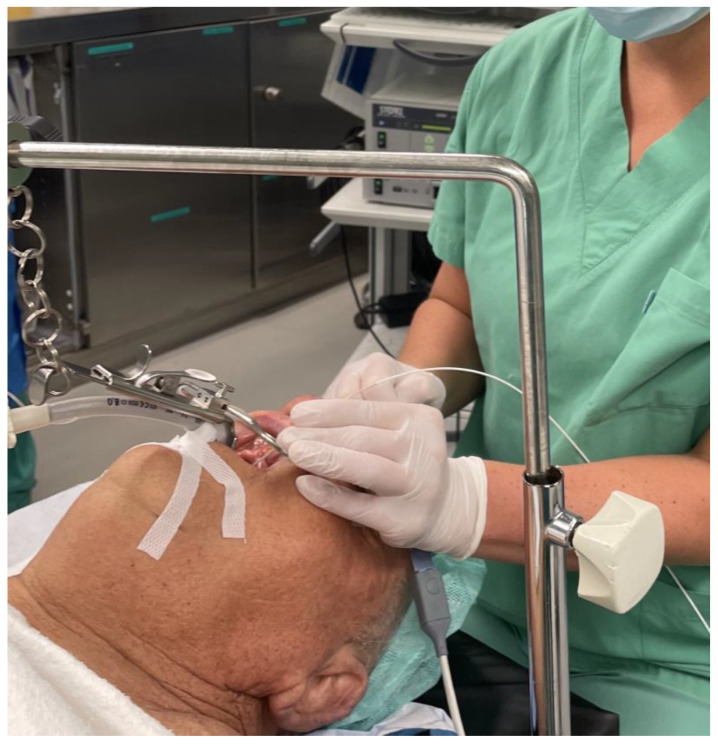
Operative setting: Examination of an oropharyngeal squamous cell carcinoma of the soft palate with the confocal laser probe.

**Figure 2 diagnostics-13-03081-f002:**
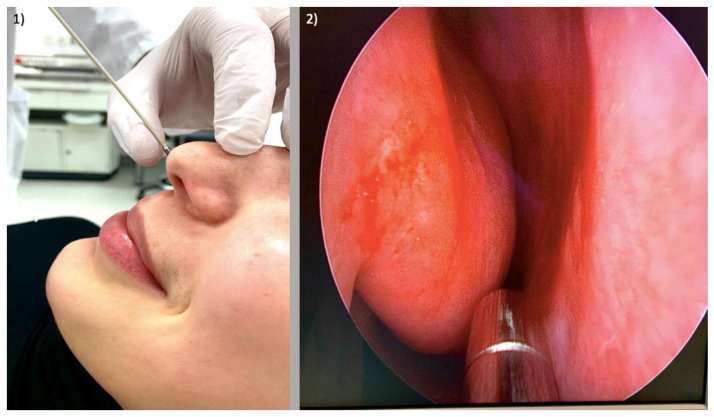
Setting for functional evaluation of allergic rhinitis: (**1**) Endonasal placement of laser probe; (**2**) Endoscopic view with laser probe between nasal septum and right inferior turbinate.

**Figure 3 diagnostics-13-03081-f003:**
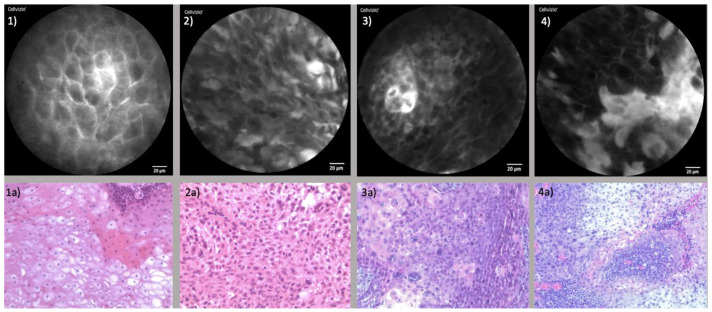
Comparison of CLE and Histopathological sections: (**1**) CLE image of healthy endonasal epithelium; (**1a**) Histopathological cross-section of regular endonasal squamous epithelium; (**2**) CLE image of endonasal squamous cell carcinoma (ESCC); (**2a**) Corresponding histopathological section of ESCC; (**3**) CLE image of oropharyngeal squamous cell carcinoma (OPSCC); (**3a**) Corresponding histopathological section of OPSCC; (**4**) CLE image of oral cavity squamous cell carcinoma (OCSCC) of the lateral tongue; (**4a**) Corresponding histopathological section of OCSCC.

**Figure 4 diagnostics-13-03081-f004:**
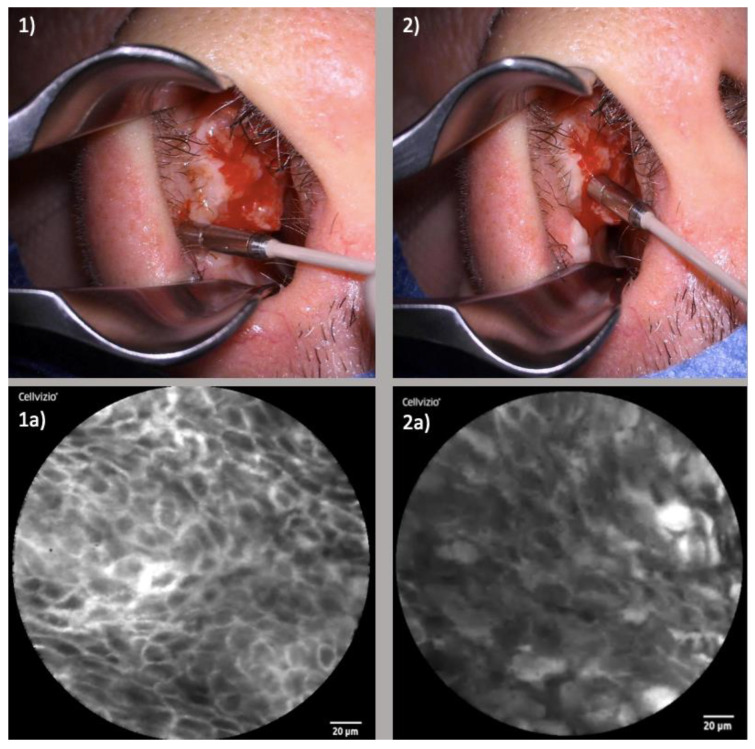
Examination of ESCC of the right nasal vestibule: (**1**) Laser probe examining the healthy tissue directly next to the tumor margin; (**1a**) Corresponding CLE image displaying regular cellular architecture, clear cell margins and homogenous uptake of fluorescein; (**2**) Examination of the tumor center; (**2a**) Corresponding CLE image with blurred, partly extinguished cell borders, irregular cell configuration and inhomogeneous distribution of fluorescein.

**Figure 5 diagnostics-13-03081-f005:**
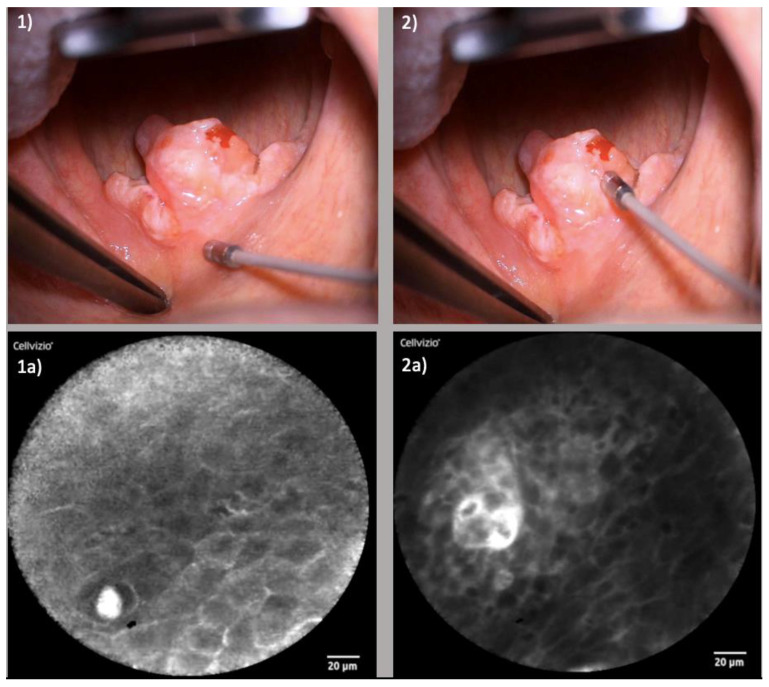
Examination of OPSCC of the soft palate: (**1**) Laser probe examining the healthy tissue of the soft palate above the tumor; (**1a**) Corresponding CLE image displaying regular cellular architecture, clear cell margins and the cross-section of a capillary; (**2**) Examination of the tumor center of the uvula; (**2a**) Corresponding CLE image presenting enlarged capillary with perivascular fluorescein leakage and irregular cellular architecture.

**Figure 6 diagnostics-13-03081-f006:**
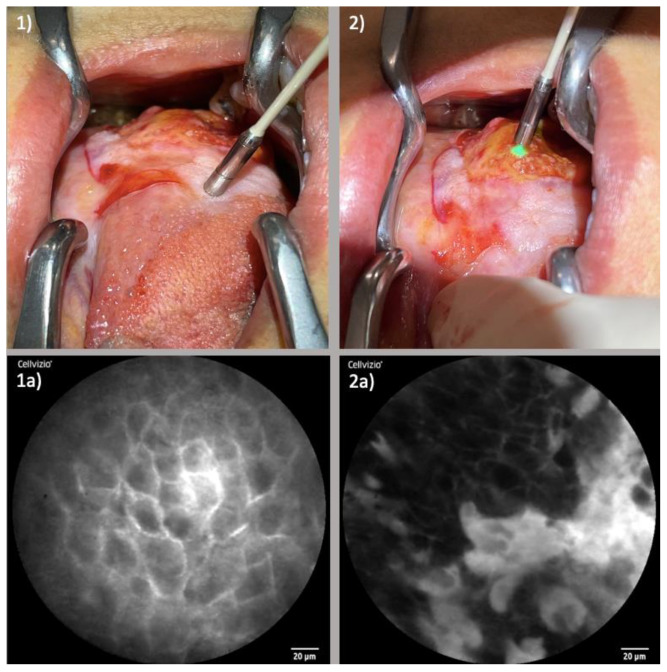
Examination of OCSCC of the lateral tongue: (**1**) Laser probe examining the tumor margins; (**1a**) Corresponding CLE image presenting neatly organized cells with clear cellular margins; (**2**) Examination of the tumor center; (**2a**) Corresponding CLE image displaying a large amount of fluorescein leaking cells with partly destroyed cell borders and areas with distinctly reduced fluorescein uptake.

**Figure 7 diagnostics-13-03081-f007:**
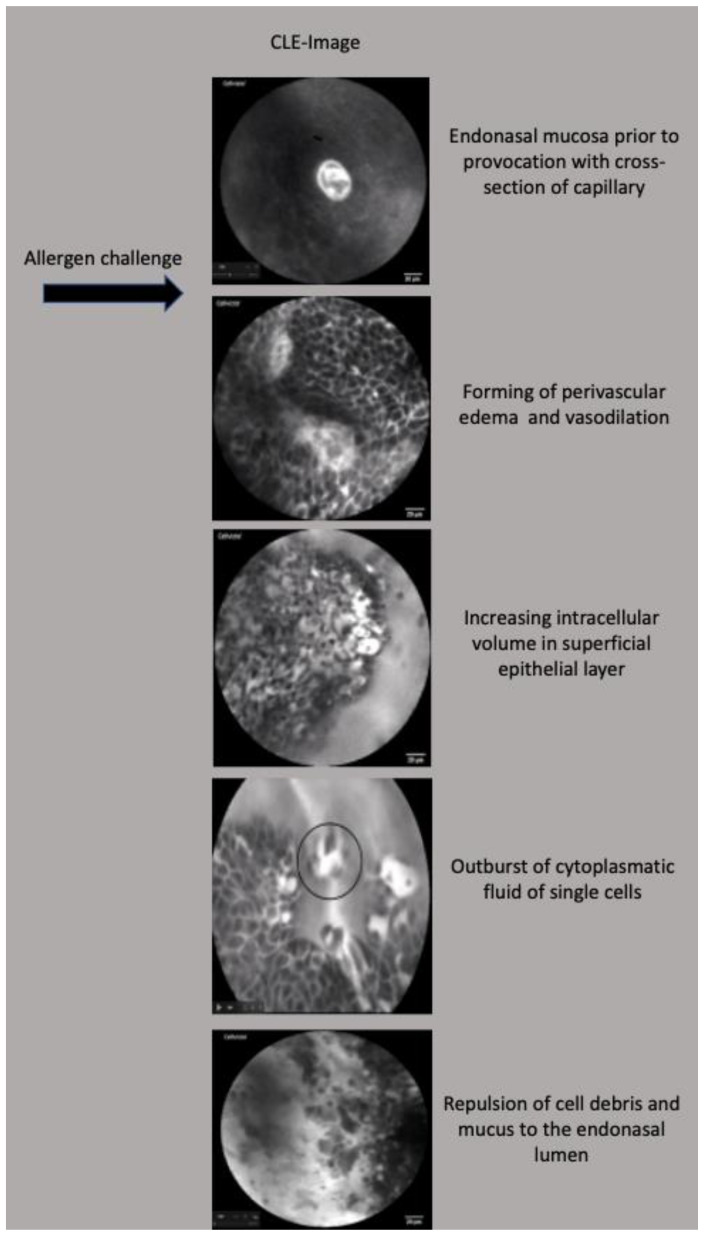
Morphological alterations in endonasal mucosa during nasal provocation testing visualized by CLE, scales in this figure are all 20 μm.

**Table 1 diagnostics-13-03081-t001:** Morphological criteria in the evaluation of mucosal tissue using CLE.

	Healthy Mucosa	Carcinoma
Cellular Configuration	Regular, oval to polygonal architecture	Deformation of cellular shape
	Cleary displayed boarders	Blurry to extinct boarders
Capillaries	Small, round to oval shaped Individually displayable	Irregular shape Formation of vascular clusters
	Absence of perivascular fluorescein leakage	Perivascular fluorescein leakage
Fluorescein Uptake	Homogenous distribution	Inhomogeneous distribution with Brighter aspects in areas of leakage
		Darker areas with reduced uptake

## Data Availability

The data presented in this study is contained within the article.

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
