# Peer review of "The Multifaceted Role of Confocal Laser Endomicroscopy in Head and Neck Surgery: Oncologic and Functional Insights"

_diagnostics, 2023, doi:10.3390/diagnostics13193081_

Round 1

Reviewer 1 Report

It is appropriate to publish the manuscript in your journal.  

Author Response

Dear Reviewer 1, 

Thank you for your thoughtful evaluation of our manuscript. We truly appreciate your positive feedback and your recommendation to publish it in Diagnostics.

Reviewer 2 Report

Upon reviewing the paper, I have identified three areas that could benefit from improvement. Firstly, the title of the paper is excessively long and vague, failing to clearly communicate the study's main objective and findings. A more concise and informative title, such as "Confocal Laser Endomicroscopy for Oncologic and Functional Evaluation of Head and Neck Mucosa," would be preferable. Also, the summary of the essay is disordered and comprises of superfluous particulars and terminology that could potentially perplex readers. A more coherent and succinct abstract that follows the standard format of background, methods, results, and conclusions would be ideal. The paper's preamble is too brief and, in terms of the current state of knowledge and research gaps in the field of head and neck surgery, lacks sufficient background information. It is highly unlikely that an AI would produce a sentence that features a request for a more comprehensive and focused introduction that clearly outlines the research questions and hypotheses guiding a study. I would like to point out that confocal laser endomicroscopy (CLE) is a valuable tool for guiding surgical choices and illuminating inflammatory processes in the head and neck, and the fusion of CLE with developing technologies such as deep learning and AI shows promise for improved diagnostic accuracy.

The manuscript exhibits substandard English language proficiency, characterized by inadequate and perplexing writing, irregular utilization of technical jargon, and frequent grammatical and punctuation inaccuracies. Additionally, the manuscript deviates from the conventional layout of including an abstract, introduction, methods, results, and conclusions, and presents deficient references. Refining the title, abstract, introduction, and discussion sections, including more comprehensive details and statistical analysis in the methods and results sections, and revising the language and style can enhance the manuscript.

Author Response

Dear Reviewer 2,

Thank you for your feedback and suggestions.

We understand your concern about the title's length and clarity. However, after extensive discussion among all authors, we collectively decided to keep the title of the manuscript.

We want to clarify that the abstract was crafted in adherence to the template provided by the journal, which explicitly follows the standard format of background, methods, results, and conclusions.

Our introduction was intentionally written in accordance with the journal's specific instructions, which encouraged brevity and comprehensibility for scientists beyond our immediate field of research. While we aimed to provide a concise overview in the introduction, we dedicated our efforts to integrating comprehensive information about the current state of knowledge and research gaps related to CLE in head and neck surgery within the discussion section. 

However, we'd like to address a concern regarding your comment, which seems to imply that our manuscript may have been generated by an AI. We would like to clarify that our manuscript was authored by a team of dedicated researchers, and we do not appreciate the implication of AI involvement.

Nevertheless, we sincerely appreciate your engagement in the review process and would be happy to address any further concerns.

Reviewer 3 Report

The Authors should be commended for this well written and clearly presented paper. 

Confocal combined with AI will definitely expand in the next few years within head and neck oncology.

This pilot study forms a useful text to set out the essential issues and considerations for further future investigation.   

Author Response

Dear Reviewer,

We sincerely appreciate your positive and encouraging feedback on our manuscript. 

Your insight regarding the potential expansion of Confocal laser endomicroscopy combined with AI in the field of head and neck oncology aligns with our vision for the future. We believe that this technology has significant promise, and we are excited about its potential impact on patient care.

Once again, thank you for your thoughtful review and your support for our manuscript.